# Expression Profiling and MicroRNA Regulatory Networks of Homeobox Family Genes in Sugarcane *Saccharum spontaneum* L.

**DOI:** 10.3390/ijms23158724

**Published:** 2022-08-05

**Authors:** Yihan Li, Yongjun Wang, Xiaoxi Feng, Xiuting Hua, Meijie Dou, Wei Yao, Muqing Zhang, Jisen Zhang

**Affiliations:** 1Center for Genomics and Biotechnology, Fujian Provincial Key Laboratory of Haixia Applied Plant Systems Biology, Key Laboratory of Sugarcane Biology and Genetic Breeding, National Engineering Research Center for Sugarcane, Fujian Agriculture and Forestry University, Fuzhou 350002, China; 2Guangxi Key Lab for Sugarcane Biology, Guangxi University, Nanning 530005, China

**Keywords:** *Saccharum spontaneum*, homeobox family, phylogenetic analysis, expression analysis, microRNA regulation

## Abstract

Homeobox (HB) genes play important roles in plant growth and development processes, particularly in the formation of lateral organs. Thus, they could influence leaf morphogenesis and biomass formation in plants. However, little is known about *HBs* in sugarcane, a crucial sugar crop, due to its complex genetic background. Here, 302 allelic sequences for 104 *HBs* were identified and divided into 13 subfamilies in sugarcane *S**accharum spontaneum*. Comparative genomics revealed that whole-genome duplication (WGD)/segmental duplication significantly promoted the expansion of the HB family in *S. spontaneum*, with *SsHB26*, *SsHB63*, *SsHB64*, *SsHB65*, *SsHB67*, *SsHB95*, and *SsHB96* being retained from the evolutionary event before the divergence of dicots and monocots. Based on the analysis of transcriptome and degradome data, we speculated that *SsHB15* and *SsHB97* might play important roles in regulating sugarcane leaf morphogenesis, with miR166 and SsAGO10 being involved in the regulation of *SsHB15* expression. Moreover, subcellular localization and transcriptional activity detection assays demonstrated that these two genes, *SsHB15* and *SsHB97,* were functional transcription factors. This study demonstrated the evolutionary relationship and potential functions of *SsHB* genes and will enable the further investigation of the functional characterization and the regulatory mechanisms of *SsHBs*.

## 1. Introduction

Transcription factors (TFs) can recognize and combine with the promoter regions to regulate the expression of target genes [1,2] and then play vital roles in plant growth and development [2,3]. The homeobox (HB) gene family is a large superfamily of TFs and is widely distributed in invertebrates, vertebrates, fungi, and plants [4,5,6]. The conserved homeodomain (HD) is the distinctive feature of the HB gene family, which consists of the helix-turn-helix DNA-binding motif (60 amino acids) [7]. Based on the phylogenetic relationship and domain composition, the plant *HB* genes were classified into 14 subfamilies, including homeodomain leucine zipper (HD-ZIP) I to IV, plant zinc finger (PLINC), Wuschel-like homeobox (WOX), knotted-like homeobox (KNOX), Bel1-like homeodomain (BEL), DNA binding homeobox and different transcription factors (DDT), plant homeodomain (PHD), nodulin homeobox (NDX), luminidependens homeobox (LD), plant interactor homeobox (PINTOX), and SAWADEE [6,8,9].

As TFs, plant *HB* genes participate in the regulation of plant growth and developmental processes [10,11,12,13]. Within the *HB* gene group, the members of different subfamilies assume different functions in plant stem cell control, organ formation, phytohormone signaling, stress response, and epigenetic regulation [14,15,16,17,18]. For example, the members of HD-ZIP III, WOX, and KNOX were involved in regulating the development of meristems [19,20,21], with the members of HD-ZIP III appearing to have a role in establishing abaxial–adaxial polarity in the embryo [20]. In contrast, WOX proteins play an important role in determining embryonic patterns, while KNOX proteins are involved in controlling the balance between meristematic and determinate growth during plant development [19,21,22]. The members of PLINC, LD, DDT, and BEL all appear to be essential for the regulation of flower development [15,23,24,25], with the members of DDT playing a role in stamen filament elongation and LD proteins mainly regulating flowering time [24,25]. Additionally, BEL proteins not only function in the regulation of fruit development but also interact with KNOX proteins and thus play overlapping regulatory roles with the NDX proteins in hormone signaling [15,16,22,26,27]. Many members of HD-ZIP II are reported to be involved in the shade response [28], while the members of SAWADEE and PHD are primarily associated with epigenetic regulation, involved in DNA methylation and histone modifications, respectively [29,30]. The findings above suggest that *HB* genes are functionally diverse in different plants. However, the functions and regulatory mechanisms of *HB* genes in sugarcane are still largely unknown.

Sugarcane (*Saccharum* spp.) is a typical C_4_ plant with a high photosynthetic rate, and it is also an extremely important sugar-energy crop. Interestingly, the two founding *Saccharum* species of the modern sugarcane cultivars, *Saccharum spontaneum* and *Saccharum officinarum*, have significantly different biological and morphological characteristics [31]. Among them, *S. spontaneum* shows a thin stem, narrow leaves, low sugar content, and high stress resistance, while *S. officinarum* is characterized by a thick stem, broad leaf, high sugar content, and low stress resistance [32,33]. *HB* genes are mainly reported to be involved in the proliferation and differentiation of the apical meristem and the morphogenesis of plants [19,20,21,22,34]. As the principal organ for photosynthesis, the leaf morphology directly affects the yield and quality of crops [35]. Therefore, we focused on exploring the potential role of *HB* genes in causing morphological differences in leaves between the two founding *Saccharum* species. In this study, we identified the *HB* genes of *S. spontaneum* based on a comprehensive comparative genomic method and explored the potential functions of *HB* genes in sugarcane based on the RNA sequencing (RNA-seq) data. Simultaneously, the microRNA (miRNA) regulatory network of *SsHBs* expression was constructed based on microRNA sequencing (miRNA-seq) and degradome sequencing. Those analyses provide a theoretical basis for further exploration of the functional characterization of *HB* genes and can provide potential gene resources for genetic improvement and molecular breeding of sugarcane.

## 2. Results

### 2.1. Genome-Wide Identification of HB Genes in S. spontaneum

Combining the results of the BLASTP program and the HMMER search tool, we initially obtained 357 gene sequences in the haplotype-resolved genome of *S. spontaneum* Ap85-441 [33]. A further determination of the presence of the HD resulted in a total of 302 sequences that were considered to be *HB* genes. After completing the allelic analysis, we named these *SsHBs SsHB1-SsHB104* based on the position of the representative alleles on the chromosome (Appendix A).

The prediction of subcellular localization indicated that all the *SsHB* genes were located in the nucleus with the exception of *SsHB81*, which was located in the chloroplast (Appendix A). The lengths of the proteins of the putative *SsHBs* ranged from 96 to 1809 amino acids, and the molecular weights of the proteins ranged from 10,094 to 201,643 Da. The theoretical isoelectric points of those proteins ranged from 3.55 to 12.68, which is similar to *Sorghum bicolor* (Appendix A).

### 2.2. Chromosome Distribution and Synteny Analysis of SsHBs

Information on the genomic location of *SsHBs* showed that 104 members were unevenly distributed on 8 chromosomes of *S. spontaneum*. In total, there were 31 *SsHBs* (accounting for 29.8% of the total number) located on chromosome 1 of *S. spontaneum*. The fewest *HB* genes were distributed on chromosome 7, with a total of 4 *SsHBs* (Figure 1A, Table 1).

In this study, we identified 344 *HB* synteny gene pairs in *S. spontaneum*, including 134 pairs of non-alleles and 210 pairs of alleles (Figure 1A, Appendix A). The results showed that the *SsHB* alleles shared high levels of collinearity among the homologous chromosomes (Figure 1A). In addition, the interspecies synteny analysis between *S. spontaneum* and *S. bicolor* showed that the position of *HB* genes changed with the breakage and recombination of chromosome 5 and chromosome 8 in *S. spontaneum* and did not cause significant gene loss (Figure 1B, Table 1). Additionally, unlike *Arabidopsis thaliana* (24.55%) and *S. bicolor* (30.61%), there were fewer *HB* genes derived from dispersed duplication in *S. spontaneum* (4.30%). It is important to point out that 54.55%, 46.94%, and 85.76% of the *HB* genes of *A. thaliana*, *S. bicolor*, and *S. spontaneum*, respectively, were derived from whole-genome duplication (WGD) or segmental duplication (Figure 1C and Appendix A). This indicated that WGD or segmental duplication events had made a valuable contribution to the expansion of the HB family in *S. spontaneum*.

### 2.3. Classification and Phylogenetic Analysis of SsHBs

The phylogenetic analysis of *SsHBs* was carried out by constructing an unrooted tree using the neighbor-joining (NJ) method. The 104 *SsHBs* were classified into 13 subfamilies, of which the HD–ZIP subfamily was composed of HD–ZIP I-IV with the largest number of HB members, and the PINTOX subfamily had only one member (Figure 2). Meanwhile, an additional phylogenetic tree was constructed using the maximum likelihood (ML) method, which is consistent with the phylogenetic relationship with the NJ tree (Appendix A). We further selected 11 species representing 5 lineages to analyze the evolutionary relationship of the HB family in *S. spontaneum*. A total of 831 HB family members were obtained from 11 species (Figure 3) and could further be divided into 14 subfamilies based on their phylogenetic relationships. Among them, *S. spontaneum* has the fourth-largest number of *HB* genes in representative species. Based on our identification and classification methods, the members of LD were not identified in *S. spontaneum* and *S. bicolor*. The *HBs* of *Selaginella moellendorffii* and *Amborella trichopoda* were distributed in 14 subfamilies, suggesting that each subfamily had a common ancestor before the divergence of angiosperm. Unicellular algae (*Chlamydomonas reinhardtii* and *Dunaliella salina*) include the members of KNOX, BEL, DDT, and PINTOX, revealing that these four subfamilies were the four ancestral classes of the HB family. Comparative genomic analysis showed that *S. moellendorffii* and *A. trichopoda* contain 39 and 55 HB members, respectively. In comparison, other eudicots and monocots have 80–148 HB members, indicating that the HB family expanded significantly after WGD. Data analysis revealed that WGD had a greater effect on the expansion of HD–ZIPI (1.0–5.0-fold increase in number of members), HD–ZIPII (2.0–7.5-fold increase in number of members) and BEL (1.5–7.5-fold increase in number of members) in angiosperms compared with that of *S. moellendorffii*, which had a limited impact (no increase) on LD and PINTOX. In addition, more members of HD-ZIP I-IV and DDT were identified in *Oryza sativa*, *Zea mays*, *S. bicolor*, and *S. spontaneum* than in *Ananas comosus*, indicating that the pan-grass ρWGD events contributed greatly to the expansion of these five subfamilies in monocotyledon lines.

To further clarify the evolutionary relationships of the HB subfamilies in *S. spontaneum*, we performed a phylogenetic analysis of the subfamilies of *S. spontaneum* and six other species. We used *C. reinhardtii* as an outgroup, and the results showed that the members of each subfamily other than the 4 small subfamilies (PINTOX, LD, NDX, and SAWADEE) could be grouped into 2–5 clades (Appendix A). In the BEL subfamily, 67 members were divided into 5 clades. *SsHB26* and *SbHB8* in Clade 5 were clustered with the outgroup, suggesting that these two genes were retained from the last common ancestor of BELs in angiosperm. Both *S. bicolor* and *S. spontaneum* in Clade 4 had two members, while *A. comosus* only contained one BEL, which may have been due to an expansion event caused by ρWGD. In the HD–ZIP II subfamily, 59 members were clustered into 5 clades, with Clade 1 containing only the HD-ZIP II members from monocots and the other clades containing family members from both monocots and dicots. These results suggested that the members in Clade 1 of HD–ZIP II (*SsHB63*, *SsHB64*, *SsHB65*, *SsHB67*, *SsHB95*, and *SsHB96*) were retained before the divergence of monocots and dicots. In Clade 2 and Clade 3, the orthologs in HD–ZIP II between *S. bicolor* and *S. spontaneum* showed a one-to-one phylogenetic relationship, indicating the conservation after the divergence of the two species. In addition, there were several segmental duplication events that occurred in SAWADEE, NDX, Clade 1 of HD–ZIP II, Clade 1 of HD–ZIP III, Clade 1 of WOX, and Clade 3 of PLINC members after the split between *S. bicolor* and *S. spontaneum*. For example, *SbHB56*, *SbHB53*, *SbHB83*, *SbHB36*, *SbHB90*, *SbHB27*, and *SbHB65* were phylogenetically distributed with two or three orthologs in *S. spontaneum*.

To investigate the characteristics of HD in SsHBs, we performed alignments for the HD protein sequences from *A. thaliana*, *S. bicolor*, and *S. spontaneum* (Appendix A). The result demonstrated that the 13 identified classes have specific homeodomain characteristics, and the highly conserved amino acid sites (Leu, Trp, Phe, Asn, Arg) [36] could still be detected. These results further supported the phylogenetic relationships among *SsHBs.*

### 2.4. Comparative Analysis between SsHBs and SbHBs

The exon/intron structures of 104 *SsHBs* and 98 *SbHBs* were analyzed (Appendix A), and the results showed that *HB* genes with 1–5 exons were the major group (64.42% and 64.29%, respectively) in *S. spontaneum* and *S. bicolor*, with the intron distributions of each subfamily being roughly similar between these two species. Most members of the HD–ZIP I, HD-ZIP II, SAWADEE, WOX, KNOX, PINTOX, BEL, and PLINC subfamilies had fewer exons (<8), and single exon genes were mainly distributed in the PLINC subfamily. In contrast, the members of other subfamilies (HD–ZIP III, HD–ZIP IV, PHD, NDX, DDT) had more exons. In addition, variations in exon structure occurred in the homologous genes of *S. spontaneum* and *S. bicolor*, such as *SsHB65*, *SsHB96*, *SsHB31*, *SsHB20*, and *SsHB27*, which had 8, 11, 13, 2, and 1 exons respectively. Their homologous genes, *SbHB56*, *SbHB98*, *SbHB1*, *SbHB5*, and *SbHB7*, had 4, 3, 18, 5, and 6 exons, respectively. There were no *HB* genes with 12 exons in *S. spontaneum* and no *HB* genes with 13, 15, or 16 exons in *S. bicolor* (Appendix A). The above results showed that both gains and losses of exons occurred during the evolution of the HB family after the divergence of *S. bicolor* and *S. spontaneum*.

To further analyze the conserved motifs of *SsHBs* and *SbHBs*, we prepared a schematic diagram of conserved motifs (Appendix A). A total of 14 motifs were identified and annotated (Appendix A). As shown in the diagram, HD (motif 1, motif 2, or motif 8) were detected in all *HBs*, and we observed that the motifs have similar orders in the same subfamily. Interestingly, motif 3 and motif 13 were only present in HD–ZIP III and HD–ZIP IV, respectively. In addition, to investigate the evolutionary forces acting on the *HB* genes, the nonsynonymous-to-synonymous substitution ratios (Ka/Ks) of 110 *HB* homologous gene pairs between *S. bicolor* and *S. spontaneum* were calculated. The Ka/Ks values of all gene pairs were less than 1, indicating that purifying selection was the main force for driving the evolution of *HB* genes (Figure 4). Notably, HD–ZIP III members had the smallest Ka/Ks ratio, indicating that the members of HD–ZIP III experienced stronger purifying selection pressure and had limited functional divergence after *S. spontaneum* and *S. bicolor* differentiation. Moreover, based on Ks, we estimated the divergence time among the 13 classes of the *SsHBs*. The average Ks was 0.93–3.28, and the corresponding divergence time was 75.84–268.85 Mya (Appendix A), demonstrating that the HB family is an ancient gene family.

### 2.5. The Expression Patterns of HB Genes at Different Developmental Stages, Gradient Developmental Leaf Segments, and Circadian Rhythm in S. spontaneum and S. officinarum

The functions of *HB* genes greatly affected the growth and development of plants. To explore the potential functions of the *HB* genes, we analyzed the transcriptome data of 12 samples from different tissues at different developmental stages of two founding *Saccharum* species (*S. spontaneum* and *S. officinarum*; detailed sample information is available in the methods section). A total of 10 *HB* genes showed extremely low expression (TPM < 3) between the two species, and transcripts of *HB70* and *HB78* were not detectable in 12 samples. We speculated that these two genes might be pseudogenes or could be expressed in tissues that were not analyzed. Another 48 *HB* genes were detected in all 12 tissues, and 5 of them (*HB27*, *HB43*, *HB85*, *HB97*, and *HB101*) showed the constitutive expression (TPM > 30 in all samples) of the two founding *Saccharum* species (Appendix A). In addition, *HB85* and *HB97* were highly expressed in 12 samples of the two founding *Saccharum* species (TPM > 50). The expression of *HB* genes in leaves at different developmental stages of the two founding *Saccharum* species showed that the expression levels of nine *HBs* (*HB3*, *HB18*, *HB23*, *HB34*, *HB62*, *HB75*, *HB97*, *HB101*, *HB102*) increased with the maturity of leaves. *HB15*, *HB85,* and *HB103* showed the opposite trend. In addition, the expression levels of six *HB* genes (*HB3*, *HB18*, *HB27*, *HB31*, *HB43*, and *HB102*) increased from the top to the bottom of the stem. However, the expression pattern of *HB36* was the opposite, indicating that these genes may participate in the transport and accumulation of sucrose (Figure 5A).

To clarify the functional divergence of the *HB* genes for leaf development and photosynthesis, we analyzed the leaf developmental gradient. We divided the 15 cm-long seedling leaves of 2 founding *Saccharum* species into 4 regions including 15 leaf segments according to previous studies [37]. They were the basal region (0–3 cm, sink tissue), transition region (3–6 cm, source-sink transition region), maturing region (6–10 cm, immature source tissue), and mature region (10–15 cm, fully differentiated and active C_4_ photosynthetic region). The analysis of the expression of *HB* genes in each leaf segment showed that the expression levels of 16 *HB* genes were extremely low (TPM < 3) in all leaf segments of two founding *Saccharum* species, and no transcript was detected for *HB78* (Appendix A). *HB61*, *HB85*, *HB97*, and *HB101* were highly expressed in each leaf segment. Moreover, the expression of eight *HB* genes (*HB6*, *HB11*, *HB15*, *HB34*, *HB60*, *HB68*, *HB81*, *HB88*) decreased from the base to the tip of leaves, while *HB12*, *HB39*, and *HB85* showed the opposite expression trend. The *HB2*, *HB7*, and *HB102* genes were highly expressed in the source-sink transition region. The expression levels of *HB17*, *HB37*, *HB44*, *HB94*, and *HB96* were significantly different among the two *Saccharum* species (high expression in one species with no or low expression in another) (Figure 5B). The above results indicated that the *HB* genes had roles in leaf development and photosynthesis.

To explore the day–night rhythm expression pattern of the *HB* genes, the leaves of two founding *Saccharum* species were collected at different time points (sampling 2 h intervals on the first day and 4 h intervals on the second day) for transcriptome analysis. Transcripts of seven *HB* genes at all-time points were not detected, indicating that these genes contributed little to the response of the sugarcane to circadian rhythm (Appendix A). The expression patterns of *HB17*, *HB26*, *HB38*, *HB44*, *HB51*, *HB94*, and *HB96* were quite different between the two founding *Saccharum* species. *HB3*, *HB27*, *HB36*, *HB61*, *HB85*, and *HB97* were highly expressed at all time points. In addition, *HB15*, *HB31*, *HB61*, and *HB97* respond to day–night transformation and display a circadian expression pattern. Among them, the expression levels of *HB31*, *HB61*, and *HB97* seem to have a certain correlation with light intensity and have the lowest expression level at noon. In contrast, *HB15* has different expression patterns: the expression level after sunrise has a short rise, and then continues to decline, and after sunset showed a rising trend (Figure 5C).

Of significance, both *HB15* and *HB97* displayed circadian expression characteristics, and the expression of *HB15* in leaves at different developmental stages was significantly different between the two founding *Saccharum* species; expression was high at the base of leaves. *HB97* had high expression levels in all tissues at different developmental stages but had peak expression levels in leaf tissues (Figure 5D). These results suggested that these two genes may play a role in sugarcane growth and development, especially in leaf morphogenesis.

### 2.6. The Regulation of miRNAs in SsHBs Expression

MicroRNAs (miRNAs) are short, endogenous noncoding RNAs that control gene expression after transcription via translational inhibition or target transcript cleavage [38,39]. Here, miRNA sequencing and degradome sequencing were used to investigate the regulation of miRNAs in the expression of *SsHBs*. Three biological replicates were set up for each sample, and a total of twelve sRNA libraries from four distinct leaf segments of *S. spontaneum* were constructed and sequenced using an Illumina HiSeq 2500 platform. After quality control and the trimming of the adaptor sequences, a total of 59,279,397 distinct reads were obtained, 24,676,472 (41.6%) of which could be mapped to the sugarcane genome (Appendix A).

To identify the known miRNAs in *S. spontaneum*, the clean reads were aligned with known miRNAs in miRbase [40]. The predicted novel (previously unreported) miRNAs were filtered using IGV-sRNA software based on the miRNA identification criteria revised in 2018 to reduce false positives. In total, 75 known mature miRNAs of 28 miRNA families and 28 novel miRNAs were identified in *S. spontaneum* (Appendix A).

In addition, the analysis of the degradome sequencing data showed that 12 *SsHBs* were identified as target genes of 8 known miRNAs and 1 novel miRNA (Table 2). Among them, miR166seq4 could target and regulate three members of the HD–ZIP III subfamily (*SsHB15*, *SsHB42*, and *SsHB47*), and *SsHB43* was regulated by both miR171seq1 and miR171seq3. The expression patterns across the leaf gradients of miR166seq1-*SsHB14*, miR171seq1-*SsHB43*, miR171seq3-*SsHB43* and miR171seq5-*SsHB39* showed a negative correlation (Pearson’s correlation coefficient ranged from −0.45 to −0.93), while the expression patterns of the remaining nine pairs showed a positive correlation. It is worth noting that the previous studies reported that Argonaute10 (AGO10) could specifically sequester miR166/165 to ensure the normal functioning of the members of the HD–ZIP III subfamily [41]. Based on this, we analyzed the expression pattern across the leaf gradients of *AGO10* (Sspon.08G0006580) in *S. spontaneum*, showing that it was also highly expressed at the basal leaf region as *SsHB15*, *SsHB42*, *SsHB47*, and miR166seq4 (Figure 6), which explained that the expression of miR166seq4 was positively correlated with *SsHB42*, *SsHB47*, and *SsHB15* in gradient developmental leaves.

### 2.7. Verification of Transcriptome Data Using qRT-PCR

Transcriptome data from the gradient-developing leaf segments were examined by quantitative real-time PCR (qRT-PCR) in representative leaf segments (LF1, LF6, LF10, LF15). The expression levels of six *HBs* (*SsHB15*, *SsHB36*, *SsHB39*, *SsHB42*, *SsHB43*, and *SsHB97*), four miRNAs (miR166-seq4, miR171-seq1, miR171-seq3, miR171-seq5), and *AGO10* were tested. The results showed that the expression patterns of the selected genes and miRNAs exhibited a consistent trend between the transcriptome and qRT-PCR data (Figure 6).

### 2.8. SsHB15 and SsHB97 Are Located in the Nucleus and Display Transcriptional Activity

The full lengths of *SsHB15* and *SsHB97* were cloned from *S. spontaneum* SES208. *SsHB15* and *SsHB97* had 2565 bp and 939 bp open reading frames (ORF), respectively, and both had typical HD domains (Figure 7A, Appendix A). The yeast GAL4 system was used to detect the transcriptional activity of SsHB15 and SsHB97. All the transformed yeast cells could grow normally on synthetic dextrose medium lacking tryptophan and supplemented with X-α-gal (SD/- trp + X-α-gal). However, the yeast transformed with the negative control vector cannot survive on SD/- Trp + AbA^150 ng/mL^ + X-α-gal; obvious blue signals could be seen in yeast transformed with the constructed fusion plasmid and pGBKT7-53, indicating that SsHB15 and SsHB97 had transcriptional activities (Figure 7B).

To investigate whether SsHB15 and SsHB97 proteins are located in the nucleus, two fusion vectors, *35S::SsHB15-EGFP* and *35S::SsHB97-EGFP*, were constructed. These two vectors were used to transiently transform tobacco epidermal cells, and 4′,6-diamidino-2-phenylindole (DAPI) was used as a nuclear location marker. The results showed that SsHB15-EGFP and SsHB15-EGFP proteins were limited to the nucleus of tobacco epidermal cells and overlapped with the DAPI signal (Figure 7C). These results indicated that SsHB15 and SsHB97 were nuclear proteins.

## 3. Discussion

This study initially analyzed the phylogenetics, gene structure, and motifs of the HB family in *S. spontaneum*. A total of 104 *HBs* were identified in the *S. spontaneum* genome, which is the fourth-largest number of *HB* genes in the representative species (110 in *O. sativa*, 148 in *Zea mays*, and 110 in *A. thaliana*). In addition, the HD–ZIP family (HD–ZIP I-IV) had the largest number of HB members in *S. spontaneum*. HD–ZIP III had the fewest genes and the least variation in gene numbers among species, consistent with previous reports that HD–ZIP III was relatively conserved across species [42]. In agreement with the results of previous studies [6], the HB gene family could be divided into 14 subfamilies based on HD features and the phylogenetic relationships. Comparative genomics revealed that KNOX, BEL, DDT, and PINTOX might be the four oldest classes among the HB family. In contrast, no LD family members were identified in *S. bicolor* and *S. spontaneum*, which could be due to a small-scale gene loss event that occurred during the evolutionary process after the divergence from *Z. mays*. During the evolution of angiosperms, WGD events are the main force leading to the expansion of TF families [43,44]. It has been shown that 90% of the increase in TFs in *A. thaliana* over the last 350 Mya was directly caused by three WGD events [43]. Our results also indicated that WGD was a driving force in the expansion of the HB gene family in *S. spontaneum*, which is in agreement with a previous study [45]. Also consistent with the results of previous studies is that higher plants contain more *HBs*, suggesting that *HB* genes may have played roles in the evolution of plants from unicellular to more complex multicellular forms [4]. The results of estimating the divergence time among subfamilies suggest that the HB family is an ancient family. This view was also well supported by the distribution of HB family members in 14 subfamilies of the *S. moellendorffii* and *A. trichopoda*.

Furthermore, the results of conserved motif analysis showed that HB members within the same subfamily have similar conserved motif distribution patterns. Moreover, these motif structures are highly conserved during evolution, which may be another reason why different subfamily members can perform specific functions. For example, motif 3 and motif 13 are only present in the HD–ZIP III and HD–ZIP IV subfamilies, respectively. This is consistent with the report that HD–ZIP family members mediate transcription in a steroid-dependent manner with the help of the START domain [46]. In addition, the number of exons was similar among members of the same subfamily between *S. bicolor* and *S. spontaneum*. The members of the HD–ZIP I family all have 1–4 exons, and HD–ZIP III family members contained more exons compared with the members of HD–ZIP I, HD–ZIP II, and HD–ZIP IV, which is relatively consistent with the findings in *P. edulis* and *V. vinifera* [9,36]. However, there was also exon gain and loss during the evolution of sugarcane, which may have resulted from the recombination and fusion of chromosome segments during polyploidization [47].

Gene function is usually considered to be closely related to expression patterns [48]. To explore the potential function of *SsHBs*, we performed a comprehensive analysis of *HB* gene expression patterns based on large-scale expression profiles of RNA-seq data, including leaf developmental gradients, diurnal cycles, and different development stages. Previous studies had reported that *AtHB13* (a member of HD–ZIP I) and *AtWOX7* relied on sugar metabolism-related pathways to regulate plant growth and development [49,50]. In our study, *HB3*, *HB18*, *HB27*, *HB31*, *HB43*, and *HB102* showed peak expression in the top internode, and the expression level in the stem gradually increased from the top to the bottom, while *HB36* had the opposite expression trend, indicating that these genes may be involved in the process of sugar metabolism. In graminaceous plants, leaf development and photosynthetic differentiation proceeded continuously from the leaf base to tip [37,51,52]. The basal leaf region maintains the activities of basic cellular functions such as DNA synthesis and cell wall synthesis, while the tip region of leaves mainly performs photosynthesis. The present study showed that the expression levels of *HB6*, *HB11*, *HB15*, *HB34*, *HB60*, *HB68*, *HB81*, and *HB88* were the highest at the leaf base, indicating that these genes were involved in basic cellular activities and related to leaf development. In the source-sink transition region, the expression of genes related to the photosynthetic mechanism will increase significantly [37]. *HB2*, *HB7*, and *HB102* have peak expression levels in this region, indicating that these genes may play roles in the establishment of the photosynthesis system. *HB12*, *HB39*, and *HB85* were highly expressed in the mature region of the leaf, suggesting that these three genes were highly associated with photosynthesis. The expression profile analysis of circadian rhythm showed that the expression levels of *HB15*, *HB31*, *HB61*, and *HB97* fluctuated according to a diurnal cycle. Previous studies have shown that genes responding to circadian rhythms are the basis of plants’ ability to sense time changes and coordinate development and metabolic processes [53,54,55,56]. This indicates that *HB15*, *HB31*, *HB61*, and *HB97* may play a role in the maintenance of plant circadian clock stability in sugarcane. In short, the *SsHBs* expression profiles obtained in this study provided an important reference for further exploring the function of the *HB* genes in sugarcane.

Among their other functions, miRNAs can mediate sequence-specific post-transcriptional gene silencing by transcript cleavage or translational repression in plants, and a miRNA can target multiple genes and regulate multiple biological processes [57,58]. Therefore, miRNAs are considered to be genetic tools involved in plant morphogenesis and the regulation of crop agronomic traits [59]. In this study, based on the sugarcane miRNA-seq and degradome sequencing, 13 pairs of targeted regulatory relationships were obtained, including 9 miRNAs and 12 *SsHBs.* However, according to the transcriptome data analysis, miRNAs can regulate the expression of *SsHBs* in a manner beyond target transcript cleavage in *S. spontaneum*. Usually, miRNA binds to AGO protein to form an RNA-induced silencing complex that plays a role in gene expression regulation and ultimately affects plant growth and development [41]. In addition, compared with AGO1, the main contributor to miRNA-mediated silencing with catalytic activity, AGO10 without catalytic activity has stronger miRNA-binding activity. Therefore, AGO10 can usually be used as bait for miR166/165 to prevent its binding to AGO1, thereby ensuring the normal expression and SAM maintenance function of HD–ZIP III members [41], and the transcriptome data in this study support this. Finally, regarding the positive correlations between the expressions of novel23-*SsHB18*, miR167seq3-*SsHB27*, miR397seq1-*SsHB3*, miR5168-*SsHB1*, miR5168-*SsHB14*, and miR5168-*SsHB56*, we speculate that these miRNAs may have modulated gene expression through translational inhibition [38,39]. This idea needs further support with experimental data. It can be seen that miRNAs not only participate in the regulation of the expression of *SsHB* genes through transcript cleavage but they may also contain more complex regulatory mechanisms of translational inhibition and AGO protein participation.

*AtHB14* (*PHABULOSA, PHB*) and *AtKNAT4* (the greatest homology to the maize *knotted-1-like 11* protein) are members of the HD–ZIP III and KNOX families, respectively. They are reported to be involved in plant growth and developmental processes such as regulating leaf morphogenesis, vascular patterning, and the deposition of secondary cell walls in several species [60,61,62,63,64]. *SsHB15* and *SsHB97* are homologous to *AtPHB* and *AtKNAT4*, respectively, although their function in sugarcane has not been characterized; nevertheless, combined with transcriptome data analysis, we speculate that these two genes may play important roles in regulating sugarcane leaf development. Additionally, we found that *SsHB15* and *SsHB97* possessed the typical characteristics of TFs, including having transcriptional activity and being located in the nucleus. This provides significant clues and a theoretical basis for further exploration of function.

## 4. Materials and Methods

### 4.1. Plant Material, RNA Extraction and Sequencing

Two founding *Saccharum* species, *S. spontaneum* SES-208 (*Ss*, 2n = 8x = 64) and *S. officinarum* LA-Purple (*So*, 2n = 8x = 80), were used to investigate gene expression patterns. The plants were grown in a greenhouse (14:10 L/D, 30 °C L/22 °C D, and 60% relative humidity) and an experimental field of Fujian Agriculture and Forestry University (Fuzhou, China). The second leaves of 11-day-old SES208 and 15-day-old LA-Purple were collected 3 h after entering the light phase. These leaves were divided into 15 1 cm leaf segments, and 4 plant samples were collected for each leaf segment as a biological replicate. Stem and leaf tissues from seedlings (35 days old), leaf roll, leaf (the first fully expanded leaf), top internode (internode number 3, i.e., stem 3), maturing internode (internode number 6 for SES208, and internode number 9 for LA-Purple, i.e., stem 6/9), and mature internode (internode number 9 for SES208 and internode number 15 for LA-Purple, i.e., stem 9/15) at the 9-month-old pre-mature stage and 12-month-old mature stage were collected from the 2 founding *Saccharum* species. Finally, the middle 4 cm of the first leaf of the 2 founding *Saccharum* species in mature stages was collected at different time points on 2 days from March 2, 2017 (starting at 6:00, sampling every 2 h on the first day and every 4 h on the second day). All samples were set up in three biological replicates. All tissues were frozen immediately using liquid nitrogen and stored at −80 °C for RNA isolation. The details of sampling were as described in previous studies [65,66].

Total RNA was isolated from various sugarcane tissue samples using the Trizol Reagent kit (Invitrogen, Waltham, MA, USA), following the manufacturer’s protocol. The cDNA libraries for each sample were constructed using Illumina^®^ TruSeq™ RNA Sample Preparation Kit (RS-122–2001 (2), Illumina). The libraries were sequenced using an Illumina HiSeq2500 instrument with paired-end 100 bp reads (Center for Genomics and Biotechnology, FAFU, Fuzhou, China). Furthermore, the gene expression abundance was quantified by transcripts per kilobase of exon model per million mapped reads (TPM) values.

### 4.2. The Retrieval of the Homeobox Sequences

The *HB* genes of *A. thaliana*, *Z. mays*, *O. sativa*, *S. moellendorffii*, and *C. reinhardtii* were retrieved from previous reports [6]. The genomic data of *S. bicolor*, *A. comosus*, *V. vinifera*, *A. trichopoda*, and *D. salina* were obtained from Phytozome (https://phytozome.jgi.doe.gov/pz/portal.html, accessed on 4 June 2020), and the genomic data of *S. spontaneum* were generated in our own laboratory [33]. To identify the *HB* sequence, two approaches were used in this study. Firstly, the sequences of HB protein previously identified from *Arabidopsis* [6] were used as queries to perform a BLASTP program search with an e-value ≤ 1 × 10^-5^. Secondly, three hidden Markov model (HMM) profiles (homeobox domain: PF00046, ZF–HD dimerization domain: PF04770, and SAWADEE domain: PF16719) were obtained from the Pfam website (http://pfam.xfam.org/, accessed on 15 July 2020) and were used as a query to search against the all-species database with the HMMER search tool [67]. All the potential candidate *HB* sequences were then manually analyzed using the SMART database (http://smart.embl-heidelberg.de/, accessed on 19 July 2020) and the National Center for Biotechnology Information (NCBI) Conserved domain database search (cd-search) (http://www.ncbi.nlm.nih.gov/Structure/cdd/wrpsb.cgi, accessed on 19 July 2020) to validate the presence of the homeobox domain. The redundant sequences were further removed by alignment, and the remaining ones were considered putative HB genes. In addition, the numbers of amino acids and isoelectric points and the relative molecular masses of the proteins were calculated using the online ExPasy program (http://www.expasy.org/tools/, accessed on 31 July 2020). The subcellular locations of the SsHBs were predicted using the Cell-Ploc tool [68].

### 4.3. Multiple Sequence Alignments and Phylogenetic Analysis of the Sugarcane Homeobox Genes

The multiple sequence alignment tool MAFFT [69] was used for multiple protein sequence alignments. The phylogenetic tree based on the alignments was inferred using the neighbor-joining method implemented in MEGA X [70]. Bootstrap analysis was performed with 1000 replicates. Additional maximum-likelihood (ML) phylogenetic trees were constructed using iq-tree with the following parameters: iqtree -nt AUTO -bb 1000 –redo [71]. The generated trees were displayed using iTOL [72].

### 4.4. Gene Structure, Protein Conserved Motif Analysis, and Calculation of Ka/Ks

The exon–intron structures of the *HB* genes were acquired from the gff3 file, and diagrams were drawn using TBtools [73]. Conserved motifs of *HB* genes were identified using the MEME program [74] with the following parameters: optimum motif widths of 12–100 residues, maximum of 14 motifs, and the other parameters were set as default. The Easy_KaKs calculation program (https://github.com/tanger-zhang/FAFUcgb/tree/master/easy_KaKs, accessed on 9 September 2020) was used to calculate the nonsynonymous (Ka) and synonymous (Ks) substitution ratios. Meanwhile, the divergence time (T) was calculated as T = Ks/(2 × 6.1 × 10^−9^) × 10^−6^ Mya [75].

### 4.5. Gene Duplication and Synteny Analysis

The Python version of MCScan (https://github.com/tanghaibao/jcvi/wiki/MCscan-(Python-version), accessed on 3 September 2020) was used to find syntenic *HB* gene pairs between the *S. spontaneum*, *S. officinarum*, and *S. bicolor* genomes. Default parameters were used. To analyze the duplication events for each *SsHB* gene, the BLASTP program and MCScanX were used [76,77].

### 4.6. The miRNA Identification, Quantification, and Identification of Target Genes

Based on the detection by the reference genes for leaf maturation (ribulose-1, 5-bisphosphate carboxylase/oxygenase (*RBCS*), sucrose transporter 2 (*SUT2*)), and cell division (cell-cycle regulation 2 (*CYD2*)), the representative segments of the developmental leaf (LF1, LF6, LF10, LF15) were selected from the seedlings of *S. spontaneum* SES-208. The RNAs of representative leaf segments were extracted using the Trizol Reagent kit (Invitrogen, Waltham, MA, USA), and the high-quality RNAs were sent to Novogene for high-throughput small RNA and RNA deep sequencing using the Illumina HiSeq platform. The adapters and low-quality data of the original miRNA sequencing data were removed. After discarding the <16 nt and >30 nt fragments, Fast QC was used to evaluate the quality of the data to obtain clean data. miRDeep2 was used to identify candidate miRNAs, and reads per million (RPM) were used to quantify miRNA abundance. Based on the degradome sequencing platform of LC SCIENCES, the mixed-pool sequencing of the degradome was performed on roots, stems, and leaves at the seedling stage and on stems and leaves at mature stages of SES208. CleveLand4 software was used for the analysis of the degradome sequencing data and the determination of miRNA target genes. GraphPad Prism was used to calculate Pearson’s correlation coefficients for the expression patterns of the miRNAs and their target genes [78].

### 4.7. Validation of Gene and miRNA Expression Levels by qRT-PCR

To confirm the expression patterns of the representative genes, qRT-PCR analysis was performed using gene-specific primers designed by Integrated DNA Technologies (https://www.idtdna.com/pages, accessed on 28 November 2020). The cDNA synthesis was performed using a StarScript II First-strand cDNA Synthesis Kit with gDNA Remover (GenStar, Peking, China). The qRT-PCR was performed following previous reports [65,66]. Glyceraldehyde-3-phosphate dehydrogenase gene (*GAPDH*) and 25S ribosomal RNA (*25S rRNA*) were selected as reference genes [79]. The miRNA expression was normalized to *25S rRNA* and detected using the All-in-One™ miRNA qPCR Detection Kit (GeneCopoeia, Carlsbad, CA, USA). Finally, the relative expression levels for each gene and miRNA were calculated using the 2^−ΔΔCt^ method [80]. The gene-specific primers and miRNA mature sequences used for qRT-PCR are listed in Appendix A.

### 4.8. Subcellular Localization and Transcriptional Activity Detection

The full-length DNA of *SsHB15* and *SsHB97* was obtained from the cDNA of SES208 using specific primers (SsHB15F: 5′-ATGGCGATGGTGGTGGTCGG-3′, SsHB15R: 5′-CACGAAGGACCAGTTCACGAACATG-3′, SsHB97F: 5′- ATGGCGTTCAACTACCACGAC-3′, SsHB97R: 5′-GTCTAGTTTCTTGTGACCGTACCT-3′) that were integrated to the *BamH*I and *EcoR*I linearized pGBKT7 vector using In-Fusion^®^ HD Cloning Kit (639648, Takara, Beijing, China). The two fusion vectors were transformed into Y2H Gold yeast strains for transcriptional activity detection. Positive and negative controls were pGBKT7-53 and pGBKT7-lam, respectively.

The complete sequences of SsHB15 and SsHB97 coding regions without the termination codons were inserted into the pFAST-R05 vector driven by CaMV 35S promoter and carrying EGFP protein by Gateway^®^ Technology for subcellular localization. DAPI was selected as the nuclear marker signal. Two target vectors, *35S::SsHB15-EGFP* and *35S::SsHB97-EGFP*, were transformed into Agrobacterium tumefaciens cells (GV3101). The Agrobacterium culture (OD_600_ = 0.4) was treated with infection buffer (10 mM MgCl_2_, 10 mM MES, and 100 uM AS, pH 5.4) at 25 °C for 2 h before infection on the back of *N. benthamiana* leaves. The treated plants were grown in the artificial climate chamber at 25 °C for 2 days. Confocal images were captured with a Leica TCS SP8X DLS.

## 5. Conclusions

A total of 302 allelic sequences for 104 *SsHB* genes were recognized and distributed into 13 subfamilies based on their phylogenetic relationships and HD characteristics. Comparative genomic analysis showed that the members of Clade 5 of BEL and Clade 1 of HD–ZIP II were retained from the evolutionary event prior to the divergence of monocots and dicots and that segmental duplication events made a greater contribution to the expansion of the HB family in *S. spontaneum*. Additionally, *SsHB15* and *SsHB97*, which may play important roles in the leaf development of sugarcane, proved to be located in the nucleus and had transcriptional activity detected by the yeast GAL4 system. Our study performed the first identification of *HB* genes in *S. spontaneum* and also provided valuable information to facilitate the future exploration of *HBs*’ functions in sugarcane.

## Figures and Tables

**Figure 1 ijms-23-08724-f001:**
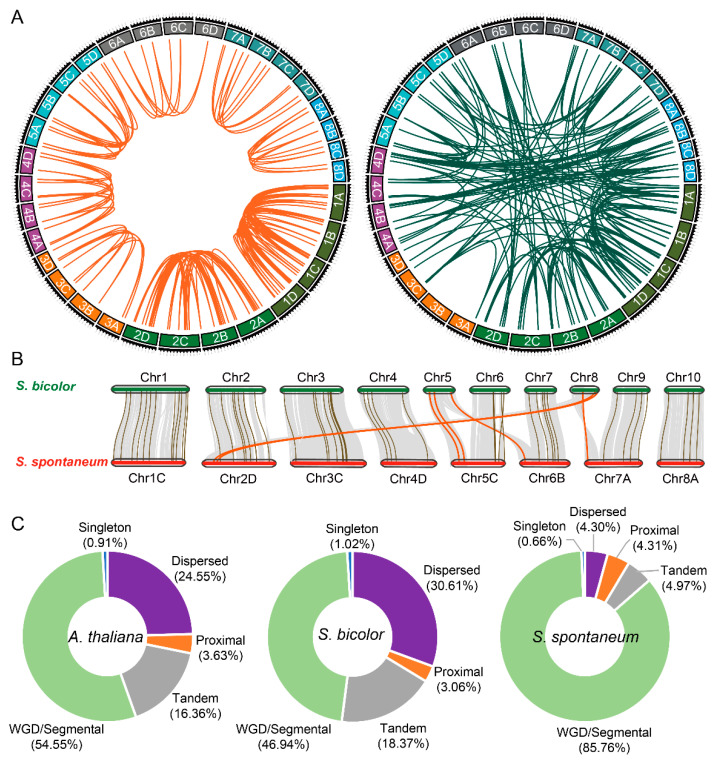
Synteny and origin analysis of *HB* genes. (**A**) Synteny analysis of alleles (left) and nonalleles (right) of *SsHBs*. (**B**) Synteny analysis of *HB* genes from *S. bicolor* and *S. spontaneum* (one set of homologous chromosomes). Gray lines in the background and brown and red lines between sugarcane and sorghum indicate the collinear blocks and syntenic *HB* pairs in normal and recombinant regions of chromosomes, respectively. (**C**) Numbers of *HB* genes from different origins in *A. thaliana*, *S. bicolor*, and *S. spontaneum*.

**Figure 2 ijms-23-08724-f002:**
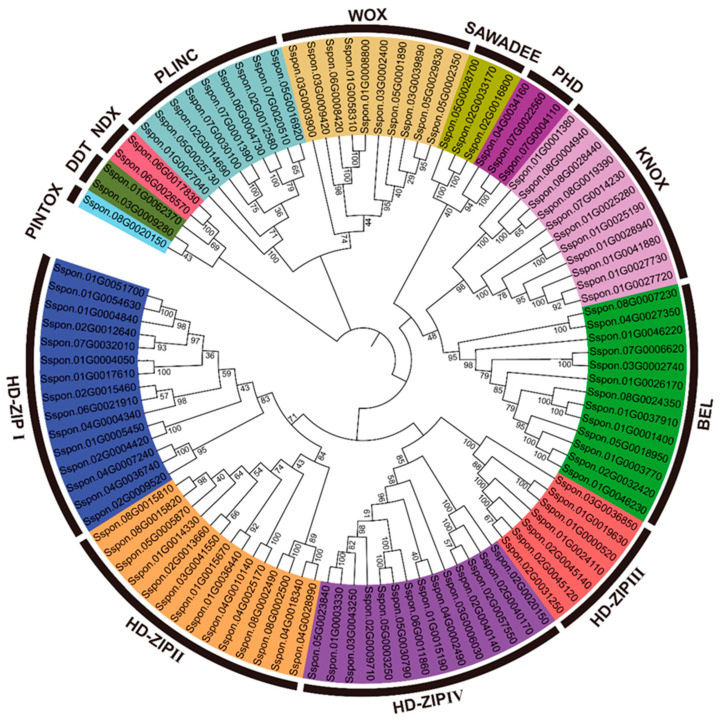
The phylogenetic tree of *SsHBs.* The labels marked in different colors represent different subfamilies.

**Figure 3 ijms-23-08724-f003:**
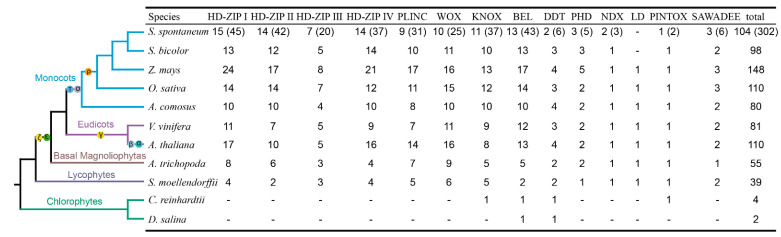
The *HBs* in 11 species among 5 lineages. The evolutionary relationships of 11 species were obtained from NCBI (https://www.ncbi.nlm.nih.gov/Taxonomy/CommonTree/wwwcmt.cgi, accessed on 4 May 2021). The numbers in parenthesis indicate the number of alleles of *SsHBs*.

**Figure 4 ijms-23-08724-f004:**
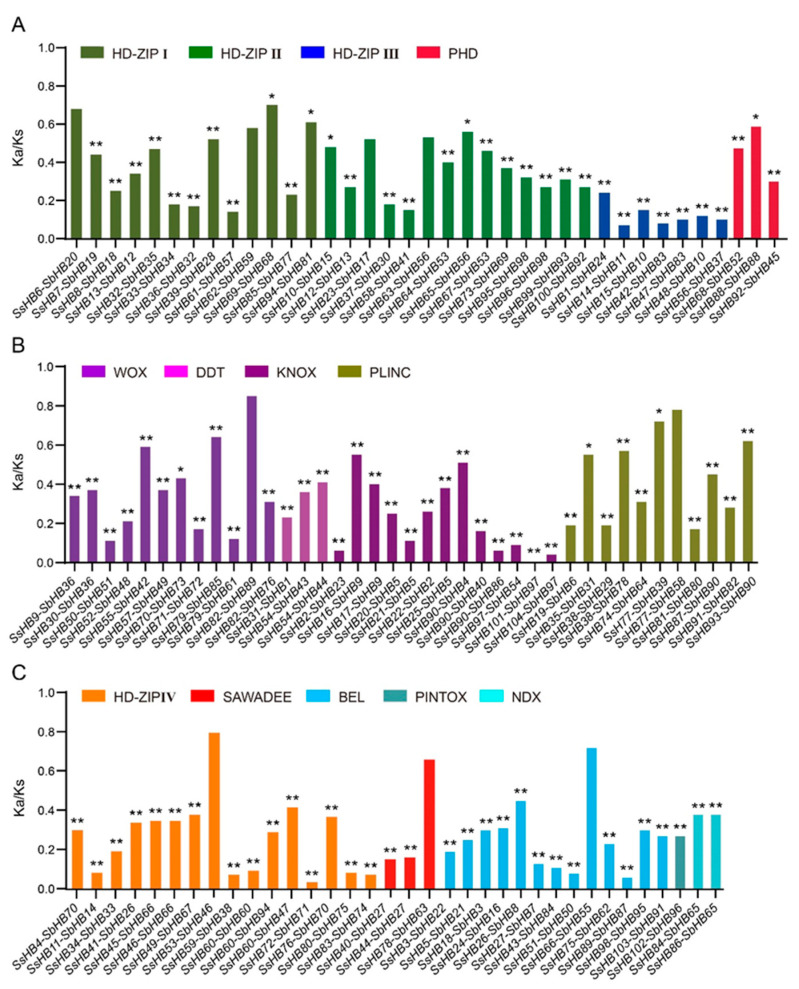
The Ka/Ks ratios of *HB* gene orthologs between *S. bicolor* and *S. spontaneum*. (**A**) The Ka/Ks of orthologs in HD-ZIP I, HD-ZIP II, HD-ZIP III and PHD subfamilies. (**B**) The Ka/Ks of orthologs in WOX, DDT, KNOX and PLINC subfamilies. (**C**) The Ka/Ks of orthologs in HD-ZIP IV, SAWADEE, BEL, PINTOX and NDX subfamilies. * indicates *p*-value < 0.05 and ** indicates *p*-value < 0.01.

**Figure 5 ijms-23-08724-f005:**
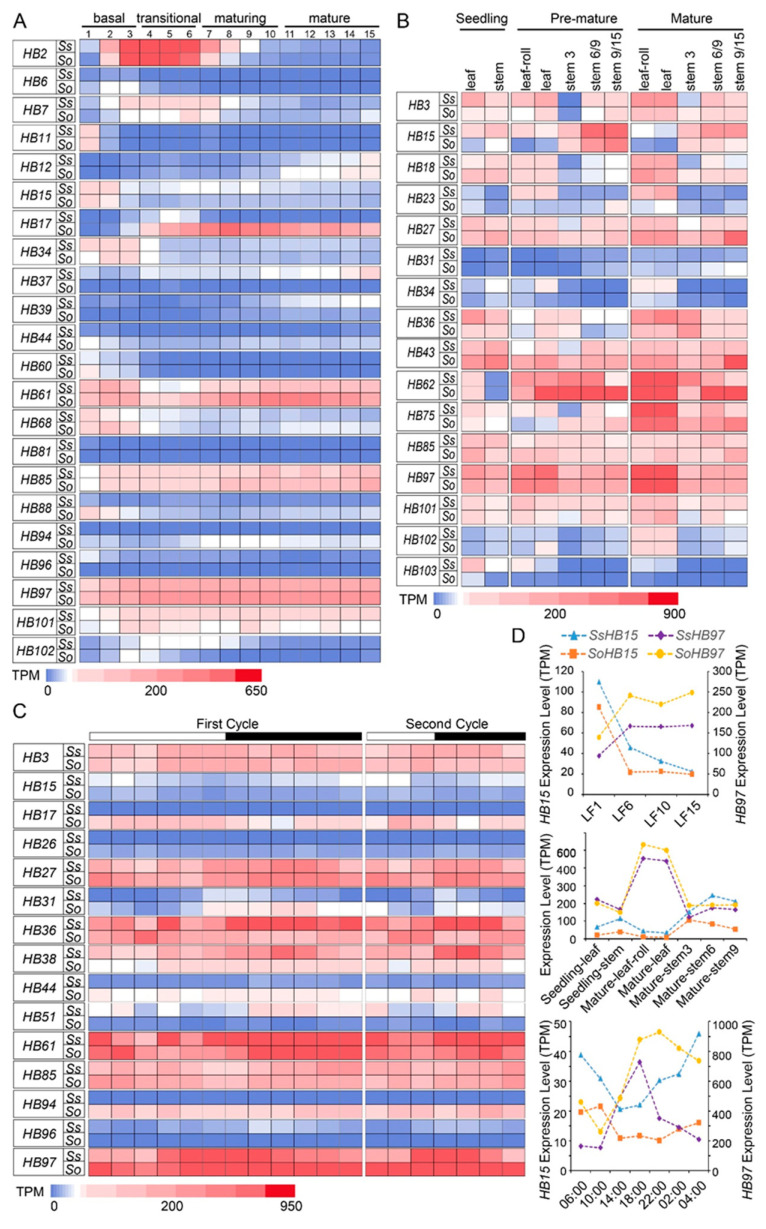
The expression pattern of *HBs* based on TPM in different tissues at different stages (**A**), gradient developmental leaves (**B**), and circadian rhythms (**C**) in 2 *Saccharum* species. The expression patterns of *HB15* and *HB97* were presented separately (**D**). Stem 6/9 indicates maturing internodes (internode number 6 for SES208 (*Ss*) and internode number 9 for LA-Purple (*So*)), and stem 9/15 indicates mature internodes (internode number 9 for SES208, and internode number 15 for LA-Purple).

**Figure 6 ijms-23-08724-f006:**
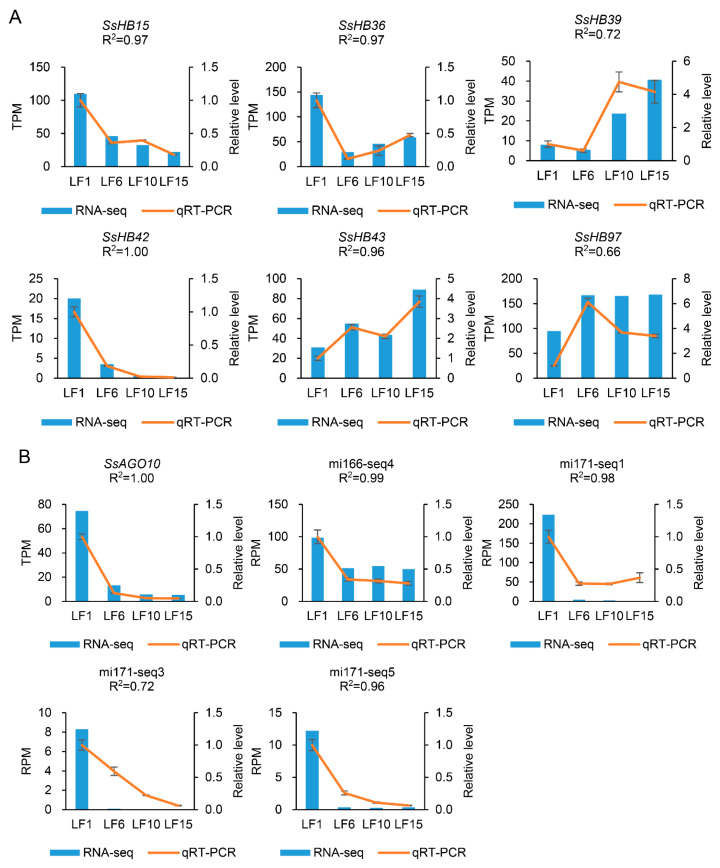
The qRT-PCR verification of *SsHBs* (**A**), *SsAGO10*, and miRNA (**B**) expression patterns in gradient-developing leaf segments. R^2^, the coefficient of determination, indicates the correlation between RNA-seq and qRT-PCR.

**Figure 7 ijms-23-08724-f007:**
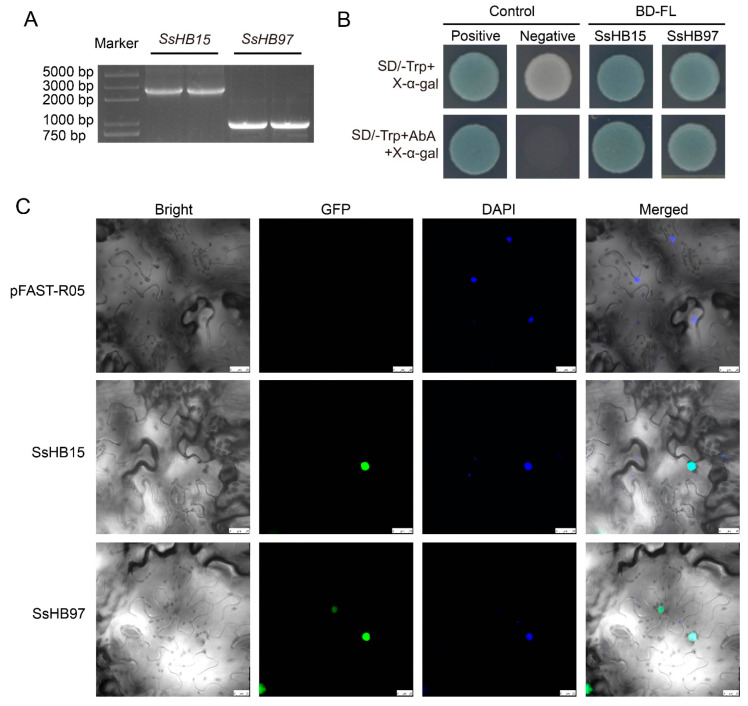
The transcriptional activity and subcellular localization of SsHB15 and SsHB97. (**A**) 1% agarose gel electrophoresis of PCR clones of SsHB15 and SsHB97 coding sequences (each sample is spread across 2 wells). The original image is shown in Appendix A. (**B**) Growth of yeast cells transformed with BD-SsHB15 and BD-SsHB97. Positive and negative controls were pGBKT7-53 and pGBKT7-lam, respectively. BD-FL indicates yeast containing the full-length target gene fused to the GAL4 DNA-binding domain. (**C**) The subcellular localization of SsHB15 and SsHB97 in tobacco epidermal cells and the white scale in each picture is 25μm.

**Table 1 ijms-23-08724-t001:** The chromosome distribution of *HB* genes in *S. bicolor* and *S. spontaneum*.

Chromosome in *S. bicolor*	Number ^2^	Corresponding Chromosome in *S. spontaneum*	Number ^2^
Chr1	24	Chr1	31 (86)
Chr2	12	Chr2	14 (46)
Chr3	15	Chr3	10 (30)
Chr4	9	Chr4	10 (33)
Chr5	5	* Chr5 ^1^	4 (11)
		* Chr6 ^1^	2 (3)
Chr6	8	Chr5	7 (21)
Chr7	8	Chr6	4 (20)
Chr8	3	* Chr2 ^1^	4 (9)
		* Chr7 ^1^	1 (3)
Chr9	6	Chr7	7 (15)
Chr10	8	Chr8	10 (25)
total	98		104 (302)

^1^ * Indicates the rearrangement region of the corresponding chromosome. ^2^ Columns 2 and 4 in the table represent the number of *HB* genes distributed on the corresponding chromosomes, and the number of alleles is in parentheses.

**Table 2 ijms-23-08724-t002:** The expression patterns of the miRNAs and their target *SsHBs* across leaf gradients.

miRNA	Expression Levels (RPM)				Target Gene	Expression Levels (TPM)				Pearson Correlation
	LF1	LF6	LF10	LF15		LF1	LF6	LF10	LF15	
novel23	1.17	46.13	50.16	8.71	*SsHB18*	30.50	117.18	118.82	89.80	0.87
miR166seq1	17.08	83.22	177.36	224.85	*SsHB14*	33.95	29.45	26.61	16.38	−0.93
miR166seq4	98.67	51.42	54.68	50.12	*SsHB15*	109.86	45.91	32.53	22.35	0.97
miR166seq4	98.67	51.42	54.68	50.12	*SsHB42*	20.02	3.49	0.57	0.13	0.98
miR166seq4	98.67	51.42	54.68	50.12	*SsHB47*	17.22	2.43	0.46	0.33	0.99
miR167seq3	6.27	49.85	93.14	109.35	*SsHB27*	16.69	46.23	60.82	86.03	0.97
miR397seq1	238.98	1209.44	974.09	1565.64	*SsHB3*	0.64	192.12	147.03	116.07	0.75
miR171seq1	223.65	4.38	3.00	0.99	*SsHB43*	30.97	55.00	43.61	89.00	−0.64
miR171seq3	8.32	0.09	0.00	0.00	*SsHB43*	30.97	55.00	43.61	89.00	−0.64
miR171seq5	12.22	0.39	0.30	0.36	*SsHB39*	8.02	5.46	23.67	40.63	−0.47
miR5168	668.85	134.94	111.77	70.96	*SsHB1*	46.97	16.34	11.53	7.71	0.99
miR5168	668.85	134.94	111.77	70.96	*SsHB14*	33.95	29.45	26.61	16.38	0.72
miR5168	668.85	134.94	111.77	70.96	*SsHB56*	11.26	7.29	2.49	1.93	0.88

## Data Availability

All the expression data are available at our lab website (http://sugarcane.zhangjisenlab.cn/sgd/html/download.html, accessed on 2 June 2021).

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
