# Peer review of "Expression Profiling and MicroRNA Regulatory Networks of Homeobox Family Genes in Sugarcane Saccharum spontaneum L."

_ijms, 2022, doi:10.3390/ijms23158724_

Round 1

Reviewer 1 Report

In the present study, the authors have identified and analyzed the genes coding for homeobox proteins in S. spontaneum.
They have also characterized their expression of HB family members to interpret the function.
The localization and transcriptional activity of two members of the HB family was also analyzed. Additionally, the study also generated information about the conserved as well as novel miRNAs targeting the HB genes. The study provides new knowledge about the HB family in 
S. spontaneum which can be explored for further functional studies. There are a few minor points that need to be addressed.

1. line 13-14: rephrase the sentence "However, little is known about HBs in sugarcane's crucial sugar crop 13 due to its complex genetic background" for clarity

2. table 1: explicitly mention in the heading row what the numbers represent in column 4 of the table represent (the number in parentheses and non-parentheses).

Authors need to check carefully grammatical mistakes

Author Response

Dear Reviewer,

Thank you for your valuable comments and suggestions, based on your suggestions we have made the following changes:

Point 1. line 13-14: rephrase the sentence "However, little is known about HBs in sugarcane's crucial sugar crop 13 due to its complex genetic background" for clarity.

Response 1. Thank you for your comments. We have modified this sentence to “However, little is known about HBs in sugarcane, a crucial sugar crop, due to its complex genetic background.” (line 13-14)

Point 2. table 1: explicitly mention in the heading row what the numbers represent in column 4 of the table represent (the number in parentheses and non-parentheses).

Response 2. As per your suggestions, we have added footnotes to Table 1 to clarify in detail the meaning of the figures in the table. (line 126-127)

Point 3. Authors need to check carefully grammatical mistakes.

Response 3. We are sorry for the shortage of writing. We have carefully checked the whole manuscript and indicated amends by using tracked changes.

Reviewer 2 Report

The manuscript, "Expression profiling and microRNA regulatory networks of Homeobox family genes in sugarcane Saccharum spontaneum L." presents the first study on HB gene family in S.spontaneum. Overall, the study would prove useful for researchers in this study area. All sections are nicely prepared quoting updated literature. The results are well-presented and justified in discussion section. A few typos/grammatical errors were spotted so please go through the text and correct them.

Author Response

Dear Reviewer,

Thank you for your valuable comments and suggestions, based on your suggestions we have made the following changes:

Point 1. A few typos/grammatical errors were spotted so please go through the text and correct them.

Response 1. We apologize for the language issues. We have improved the language and corrected the grammatical errors. Please see the revised manuscript.

Reviewer 3 Report

The manuscript of Li et al. explored the potential function of Homeobox (HB) genes in sugarcane (Saccharum spontaneum) via RNA sequencing and analysis. Unravelling these specific genes that are directly related to the yield and composition (quality) of this C4 crop is indeed highly relevant. Therefore this paper deserves recognition in this field of research. The article is well written and the applied methodology merits publication in International Journal of Molecular Sciences. After reading the manuscript I only have some minor comments which ought to be addressed before publication:

-          - In 2.1 the authors mentions “comprehensive identification method”, please refer specifically to the corresponding section in Materials and Methods.

-          - Sometimes “HB” is written in italic, sometimes it is not. Please be consistent throughout the text.

-         -  The MW, amount of amino acids, isoelectric point of proteins was calculated/estimated via a software tool (ExPasy). How accurate is this estimation/calculation? For instance, for the isoelectric point, two decimals are presented. Is the calculation that accurate? Is anything known regarding uncertainty/standard deviation ?

-       -    Numbers and text are difficult to read in Figure 3. Please increase font-size.

-          - The authors display the Pearson correlation for the expression patterns of leaf gradients. Which (statistical) program/software tool did the authors use ? Please elaborate in Materials and Methods.

-          - What does R2 mean/stand for in Figure 6A and B ? Please elaborate in the text.

Author Response

Dear Reviewer,

Thank you for your valuable comments and suggestions, based on your suggestions we have made the following changes:

Point 1. In 2.1 the authors mentions “comprehensive identification method”, please refer specifically to the corresponding section in Materials and Methods.

Response 1. Thank you very much for pointing out the shortcomings in our writing. We have revised this sentence according to the method section to “Combining the results of the BLASTP program and the HMMER search tool, we initially obtained 357 gene sequences in the haplotype-resolved genome of S. spontaneum Ap85-441.” (line 88-90)

Point 2. Sometimes “HB” is written in italic, sometimes it is not. Please be consistent throughout the text.

Response 2. Thank you for pointing out the issues, we have checked the whole manuscript. However, it should be noted that “HB” is written in italic when indicated as genes, but when indicated as proteins, we have written it as normal.

Point 3. The MW, amount of amino acids, isoelectric point of proteins was calculated/estimated via a software tool (ExPasy). How accurate is this estimation/calculation? For instance, for the isoelectric point, two decimals are presented. Is the calculation that accurate? Is anything known regarding uncertainty/standard deviation?

Response 3. Thank you for your valuable suggestions and for pointing out the issues. We should note that ExPASy is a World Wide Web server with free online access to several protein identification and analysis tools, including Compute pI/MW, ProtParam and TagIdent and others. These tools can perform analysis of physicochemical properties such as isoelectric point and molecular weight based on protein sequence. The analysis results can provide a reference for scientific research, but there are certain shortcomings and standard deviation. For example, Compute pI/MW is not sufficient for accurate calculation of pI values of basic proteins [1].

Point 4. Numbers and text are difficult to read in Figure 3. Please increase font-size.

Response 4. As per your suggestions, we have adjusted the font size in Figure 3 as large as the page allows and have changed the images in the manuscript. (line 173)

Point 5. The authors display the Pearson correlation for the expression patterns of leaf gradients. Which (statistical) program/software tool did the authors use ? Please elaborate in Materials and Methods.

Response 5. We have added the corresponding description and references in section 4.6 of Materials and Methods. (line 574-576)

Point 6. What does R2 mean/stand for in Figure 6A and B ? Please elaborate in the text.

Response 6. Thank you for pointing out the problem and letting us find our mistakes. We have redrawn Figure 6 and described the meaning of R2 in detail in the heading row. (line 350-351)

References:

  1. Wilkins MR, Gasteiger E, Bairoch A, Sanchez JC, Williams KL, Appel RD, Hochstrasser DF: Protein identification and analysis tools in the ExPASy server. Methods in molecular biology (Clifton, NJ) 1999, 112:531-552.
